# Friction Reduction by Dimple Type Textured Cylinder Liners—An Experimental Investigation

**DOI:** 10.3390/ma16020665

**Published:** 2023-01-10

**Authors:** Markus Söderfjäll, Carsten Gachot

**Affiliations:** 1Division of Machine Elements, Luleå University of Technology, 97187 Luleå, Sweden; 2Tribology Research Group, Vienna University of Technology, 1040 Vienna, Austria

**Keywords:** surface texture, cylinder liner, friction reduction, piston ring

## Abstract

Applying texture to a surface in a tribological interface will influence frictional performance, which has been investigated in several previous studies. However, since varying operating conditions heavily affect the frictional performance and optimum texture dimensions, more work in this field is required. There are few experimental studies concerning the influence of texture on friction especially under sliding lubricated conditions and even fewer at high sliding speeds. In this work, the effect of texture on frictional losses between the piston ring and cylinder liner is studied experimentally. The texture is of the dimple type, with a diameter and depth of 300 µm and 3 µm, respectively, applied to the cylinder liner surface. Experiments are performed with sliding speeds close to real piston sliding speeds. A clear reduction in frictional losses is observed with the textured cylinder liner. Moreover, qualitatively comparing the experimental results to a numerical model shows a good correlation.

## 1. Introduction

Despite the ongoing triumphal procession of e-mobility, the optimization of combustion engines and thus the reduction of fuel and oil consumption is still of high priority in the automotive industry, especially in heavy-duty equipment such as trucks and construction machinery. Mechanical friction is a large contributor to fuel consumption in those applications where a heavy-duty diesel engine (HDDE) is used. The piston and piston ring pack are responsible for approximately half of these frictional losses [1]. Moreover, the oil control ring is responsible for most of the piston and ring pack friction, making it the most interesting single component to study when optimizing for lower fuel consumption by reduced friction.

In general, there are plenty of methods available for minimizing friction and wear in the case of dry and lubricated friction situations that have already been developed in the past decades. These range from mechanical methods (grinding, honing), lithographic methods (UV or electron beam lithography), and high-performance coatings (e.g., Diamond-like carbon coatings) to texturing methods (e.g., laser surface texturing) [2]. In this context, lasers as a light source offer great potential for controlling tribological properties. Under dry friction conditions, the laser-based generation of remotely well-defined topographic patterns in the surface of the affected friction partners can provide for the storage of wear particles in the “pockets” generated in this way, thereby reducing three-body abrasive wear.

The advantage of laser technology is based on its process flexibility regarding integration into process chains and its almost universal applicability for a wide range of materials. Today, beam sources with short and ultra-short pulses in the range of a few femtoseconds exist, which allow fast and precise processing of practically all relevant material surfaces and geometries. The possible friction reduction by laser texturing surfaces is in the range of 25–50% [3].

Laser-induced structures in material surfaces also offer great relevance for the smallest amounts of lubricant. With the help of these laser-induced dimples, insufficient lubrication conditions can be avoided by reliably supplying the contact area with sufficient lubricant held within the dimples. The lubricant film thickness, which is decisive for the separation of the friction partners involved, can be precisely controlled by means of additional hydrodynamic pressure contributions in the tiny lubricant gaps of, for example, high-speed bearings. Current trends show the increased use of low-viscosity oils, e.g., 5W20, 0W20, or 0W8. Lower viscosity means that correspondingly lower shear forces are transmitted, reducing fuel consumption by a further 0.5% [4]. The problem of increased mixed or boundary lubrication in this sequence is typically mitigated by the formation of tribo-chemical reaction layers, which in turn reduce friction and wear.

In addition, laser surface texturing can be used to influence lubricant film thickness. An interesting variant of this is direct laser interference patterning (DLIP), which produces precise topographic patterns in micro- and nano-dimensions with high reproducibility. This method allows the adjustment of periodic hydrodynamic pressure distributions in micrometer dimensions and can therefore make a decisive contribution to the more precise control of lubricant film thickness in micrometer dimensions. Rosenkranz et al. were also able to demonstrate in the case of insufficient lubrication situations that laser interference structured steel surfaces exhibit an increase in lubricant film life by a factor of 130 [5]. The laser tool is characterised in particular by its high process speeds, clean process control and its universal applicability for different material surfaces. In addition, by varying the wavelength, the pulse duration (e.g., femto-, pico-, or nano-second operation), and the energy density used, the laser allows the induction of a wide variety of micro-metallurgical phenomena.

As far as cylinder liner surfaces are concerned, there are numerous research articles in this field describing the pros and cons of laser surface texturing. Ma et al. studied the effect of inclined groove textures applied to a cylinder liner surface of a diesel engine and dimples on the respective gas ring [6]. They could show that the texturing of the cylinder liner and the piston ring leads to an increased film thickness and reduced friction. Moreover, Kang et al. investigated the influence of laser-fabricated dimples with a typical depth of 8 µm near the top dead centre (TDC) and the piston skirt contacting region. Their findings highlighted improved hydrodynamic lubrication with reduced engine oil and fuel consumption [7]. Furthermore, Xu et al. studied the effect of larger dimples (500 µm diameter, 2–3 µm depth) produced by electrochemical mask etching and smaller dimples (70–80 µm diameter and 5–7 µm depth) fabricated by laser surface texturing on a) the barrel-shaped ring liner pair, b) the taper-faced ring liner pair, and c) the oil control ring liner pair. Their research work clearly indicates that larger dimples regarding the barrel-shaped ring liner pair are only beneficial for low oil temperature and under hydrodynamic conditions, whereas smaller dimples seem to work well under low oil temperature and mixed lubrication. For the taper-faced ring liner pair, smaller dimples enhance oil scraping efficiency and oil film distribution stability. However, concerning the oil control ring liner pair, only a small influence could be noticed [8]. Unlike the numerous research papers in the experimental field, only a few articles address numerical calculations, and even fewer address the combination of experimental and numerical research papers. A previous, solely numerical, investigation by the main author of this paper [9], hereafter referred to as: “the numerical study”, investigated the influence on friction of spherically shaped dimples applied to the cylinder liner. In the numerical study, dimensions of the dimples, layout, and amount of surface covered were investigated with respect to friction reduction. In the numerical study, a twin land oil control ring (TLOCR), which is the most common type of oil control ring in heavy-duty diesel engines, was modelled. More specifically, a segment of one single land of the oil control ring was modelled, thus, not considering ring gap effects or ring twist, and also assuming perfect conformability to the cylinder liner circumference. The model considered 2D fluid flow with the influence of plateau surface roughness with a mass-conserving cavitation model for the hydrodynamic lubrication and a contact-mechanics model includes mixed lubrication; moreover, the inertia of the piston ring was also considered in the numerical model.

The scope of this particular study is to experimentally measure the effect on friction with cylinder liner textures, i.e., laser-induced dimples with a diameter of around 300 µm and a depth of 3 µm, where the parameters of the texture are those defined by the previous numerical study [9]. This study, therefore, aims to both investigate the influence on frictional losses of the texture and also qualitatively compare the results from the numerical study on friction reduction to the measured friction reduction. There are clear benefits to placing the texture on the cylinder liner instead of on the piston rings. Firstly, when the piston ring slides over the texture a constant varying geometry is seen by the piston ring this means that the texture can aid in hydrodynamic pressure generation. This would not be possible if the texture is placed on the piston ring since it would then be stationary relative to the contact. Secondly, fuel consumption is affected by friction power loss and is, therefore, dependent on speed. Placing the texture on the liner, therefore, allows for only specific parts of the liner to be textured, preferably the mid-section of the liner, where the speed is greatest. This leaves the reversal zones untextured, where the friction power loss is low and the lubrication regimes are more those of mixed and boundary lubrication. This is beneficial since the risk of increased oil consumption or wear in the reversal zones caused by the texture is minimised. Previous studies performed with textured cylinder liners are insufficient because of two reasons. Firstly, the design of the texture is typically made by guesswork or what is possible to produce with a certain method of texturing. In this work, the design of the texture is based on the results from a numerical model developed especially for this purpose [9]. Secondly, the experiments were typically performed at low sliding speeds, far lower than the mid-stroke speeds of any typical engine, thus not investigating the potential for influence on fuel consumption. In this work, the experiments are performed at speeds similar to the real piston speeds of a typical combustion engine.

## 2. Method

The experimental equipment used in this study can be seen in Figure 1. The test equipment can accept standard production cylinder liners and piston rings. Friction is measured in a floating liner-type configuration, where the cylinder liner is placed on and only supported by load cells. Both the cylinder liner and lubricating oil temperature can be controlled. The cylinder liner is heated with two separate heating elements covering the circumference of the liner in two axial positions. The cylinder liner temperature is measured by thermocouples on three different axial positions with three different thermocouples at each axial position, making a total of nine thermocouples on the liner. The lubricating oil is supplied to the contact via nozzles from underneath the piston ring and its holder similar to an actual engine. The temperature of the lubricating oil is measured just before exiting the nozzle and regulated based on this measured temperature. The piston ring holder is linearly guided to avoid any piston contact and only allow for the piston rings to be in contact with the cylinder liner; thus, only piston ring friction is measured. The piston ring holder is a modified standard type of piston and can therefore hold any or all configurations of the three piston rings used in a typical HDDE. The experimental equipment is based on an inline six-cylinder engine because of the low vibration characteristics of such an engine. This allows tests to be performed at realistic piston speeds without extensive vibrations in sampled friction force. A more in-depth detailed description of the test equipment can be found in [10].

The piston ring used in this study was a standard-type twin land oil control ring with a land width of 150 µm and tangential ring tension of 33 N, resulting in a nominal contact pressure of 1.7 MPa. This oil control ring geometry and load correspond to those used in the numerical study [9].

As mentioned previously, the design and layout of the texture used in this study were based on the previous numerical modelling results in [9]. The texture was produced according to the following requirements:Shaped like spherical indents.Diameter: 300 µm.Depth: 3 µm.Amount of surface covered by dimples, i.e., area density: 33%.Zig-zag pattern (diagonal layout relative to the sliding direction), see Figure 2 for an explanatory picture.Texture only in the mid part of the liner, schematically depicted in Figure 2.

Preferably, for friction reduction, the area density should be greater according to the numerical study. However, due to limitations, this was the highest possible area density able to be produced with high accuracy at the time of this study.

A surface measurement of a single dimple can be seen in Figure 3.

The texture is placed on the mid part of the stroke so that the piston speed would be greater than 2.2 m/s at a test rig speed of 1200 RPM when the oil control ring is in contact with the textured surface. This means that for crank angles lower than 19° and greater than 107°, defining 0° crank angle at TDC, no textured surface is seen by the oil control ring. This is depicted in Figure 4 for a test rig speed of 1200 RPM.

The cylinder liner blanks used for texturing were standard-type cast iron cylinder liners with a slightly smoother surface than typical production engine liners of the current standard. In total, two different identical specimens of each liner type were used; these are referred to as reference (Ref) and texture (Txt), to differentiate between the individuals they are numbered with a 1 or 2. The cylinder liner blanks used as references were identical twins to the blanks on which textures were applied.

The lubricating oil used in the tests was a standard type of heavy-duty diesel engine oil with specification 10W30. The kinematic viscosity was measured to 78 cSt and 11.3 cSt at 40 °C and 100 °C, respectively. The order in which the tests were performed can be seen in Table 1. It should be noted that the test is performed in an A-B-A fashion, where the first and last test in the test series is performed with the same components for indicating repeatability and therefore reliability of the test method.

Keeping noise in the sampled friction signal at reasonably low levels is important, therefore, the highest appropriate running speed is 1200 RPM. With greater test speeds, resulting in increased inertia, the friction signal would contain an extensive amount of noise. The test equipment has a stroke of 90 mm, whereas the truck engine that is the focus of this study has a stroke of 160 mm. Furthermore, the truck engine typically operates around 1100 RPM; therefore, some measures are needed when setting the conditions for the test. By running the test rig at a lower temperature than typical engine temperatures, the resulting greater viscosity will simulate higher sliding speeds than the actual speed in the test equipment. To calculate the simulated speed increase in the test, a Hersey parameter is defined as: $\frac{v\cdot \mu }{N}$, where v is the speed, μ is the viscosity of the lubricant, and N is the load. With this input, an expression for calculating truck-engine speeds equivalent to the test-rig speed can be written as: $v_{truck}=\frac{v_{test\;rig}\cdot \mu_{test\;rig}}{\mu_{truck\;engine}}\;$. The load will, because of the design of a TLOCR, be the same in the test rig as in an actual engine and, therefore, vanishes from the expression. A typical heavy-duty truck engine operates with an oil temperature of 95 °C, resulting in a dynamic viscosity of 10 mPa∙s. Test parameters were selected to meet two different criteria. Firstly, a representation of the typical operating conditions for a heavy-duty diesel engine. Secondly, to retrieve data with a relatively low noise level for better visualisation when compared to the numerical study results. The test conditions used for all component combinations in this study can be seen in Table 2, the speed used for calculating the equivalent truck engine speed was the mean of the absolute value of the piston speed for the entire stroke. The sampling frequency was set to 20 kHz in all tests performed, which results in the number of sampled points per revolution being 1000, 1500 and 2000 at 1200, 900 and 600 RPM respectively. The duration of the test at each test point was 30 min.

In the numerical study [9], the dynamic viscosity of the lubricant in the model was set to 10 mPa∙s. The model solves for the effect of the texture at one defined speed at a time. The model was used to find the potential for friction reduction with textured cylinder liners at three different constant speeds, 5, 7.5 and 10 m/s, thus representing most of the engine stroke where friction power loss occurs. Results from the numerical study on potential friction benefit compared to a standard type of cylinder liner as a function of area density are depicted in Figure 5. An area density of 33% is highlighted in Figure 5 since this corresponds to the area density of the texture used in this experimental study.

Since the points investigated in the numerical study only partially cover the experimental space defined by the DOE in Table 2, some complementary points were investigated with the numerical model. These complementary points were modelled with the viscosity set to 24 mPa∙s and performed at siding speeds of 1, 3, 5 and 7 m/s, which covers the viscosity and sliding speeds defined in the DOE when the piston ring is in contact with a textured surface. These complementary points together with the original points from the numerical study can be seen in Figure 6. Figure 6 also features a second-degree polynomial fitted to the numerical results. This polynomial has the following expression: $y=ax^{2}+bx$, where a=−802.42 and b=319.24. The curve fit will be used when comparing the numerical model friction reduction prediction to the experimental results.

Friction reduction predicted by the numerical model can be imposed on each sampled data point in the experiment, thereby qualitatively comparing the numerical model to the experimental results for the portion of the stroke where texture is present. More specifically, the predicted friction reduction can be imposed on the friction measured for the reference liners. These results can then be compared to the friction measured for the textured liners, resulting in a qualitative comparison of the numerical model to the experimental results. The actual temperature of the cylinder liner in the experiments will be greater than the selected temperature due to frictional heating. A 2nd-degree polynomial can be fitted on the experimental results on cylinder liner temperature, measured at three different positions on the stroke (TDC, middle of the stroke and bottom dead centre (BDC)). The measured cylinder liner temperature is then used to calculate the viscosity of the lubricant in the experiment as a function of the crank angle degree. The piston ring speed at each measured data point can thereafter be multiplied with the corresponding calculated viscosity. By interpolation in the curve fit shown in Figure 6 at the crank angles where the piston ring is in contact with a textured surface according to Figure 4, the reduction predicted by the numerical model as a function of crank angle degree can be found. This predicted friction reduction is thereafter deducted from the friction measured for the reference liners resulting in a crank angle resolved friction force trace. This friction force is then used to qualitatively compare the numerical model to the experimental results on piston ring friction.

## 3. Results and Discussion

Run-in effects and temperature stabilisation influencing friction occur at the beginning of each test. Data from the first half of each test, the first 15 min, are, therefore, discarded to study the steady-state results. Friction is extracted by averaging all sampled crank angle resolved friction traces from the second half of the test. These friction traces for all tested cylinder liners at the different operating conditions specified in the DOE are depicted in Figure 7. For better visualisation of the crank angle resolved friction, a moving average filter is applied to remove excess noise. The filtered friction force as a function of the crank angle can be seen in Figure 8. Hereafter only the filtered friction signal is shown because of the ease of interpretation. As can be seen, the friction force is significantly reduced with the textured cylinder liners. Moreover, the repeated test with cylinder liner reference 1 shows the great repeatability of the test method, thus implying that the results are reliable.

Qualitative comparisons of the experimental results with the numerical model were made by implementing the method described earlier utilizing the relation between friction reduction and the product of viscosity and speed, depicted in Figure 6. This predicted friction reduction is depicted in Figure 9, together with filtered friction results for the reference liners and the textured liners. To aid with visualisation and give a better understanding of both the measured and numerically predicted friction reduction from the textured cylinder liners, mean curves for all tests with reference liners and textured liners respectively are depicted in Figure 10 together with the friction decrease predicted by the numerical model.

The friction reduction predicted by the numerical model is close to the experimental results at all three tested speeds. A distinct trend of a worse correlation between predicted friction reduction and measured results with increasing speed can however be found. This could be influenced by the surface roughness being slightly smoother in the experiment than in the numerical model. As a result, the lubrication regime in the experiments is more into the hydrodynamic regime. By observing the crank angle resolved friction traces, especially for the reference liners, a very distinct hydrodynamic trend can be seen around mid-stroke because of the almost sinusoidal shape of the friction trace. This makes it reasonable to assume that the lubrication regime is mostly hydrodynamic around mid-stroke, where the texture is placed, for all tested speeds. As shown in the numerical study [9], dimples themselves generate hydrodynamic pressure like a micro bearing, however, the pressure generation is less than that with a non-textured surface partially because of side leakage. This leads to somewhat reduced minimum film thickness resulting in more mixed lubrication on the non-textured plateau areas on the cylinder liner, which slightly increase the friction in these areas. However, the dimples promote hydrodynamic lubrication due to the increased film thickness over the dimples because of the geometry of the dimple itself. The material removed from the surface to create the dimple results in an enforced greater mean film thickness thus reducing the friction from a global or component perspective. The reduction in friction is achieved even though the system is transitioning more into mixed lubrication from a local perspective. The reduction in friction is, therefore, dependent on the lubrication regime in which the system is operating for the non-textured case. It is reasonable to assume that the friction decrease from surface texture is greater when the non-textured situation is more into a mixed lubrication regime. This could be part of the explanation for the greater difference in the qualitative comparison between the numerical model and the measured data at greater speeds. Nonetheless, at the lowest speed tested, 600 RPM, the numerical prediction on friction reduction is almost identical to the measured difference between the textured and non-textured cylinder liners. The predicted reduction is just in the middle between the two different measured cylinder-liner individuals, therefore, when comparing the mean curves in Figure 10, the numerical results are almost the same as the measured friction reduction for the lowest tested speed. It can be said the numerically predicted friction reduction is reasonable. However, as stated previously, because of the less rough surface used in this study compared to the numerical study, the numerical model is not validated. Nevertheless, the numerically predicted friction reduction is very close to the measured reduction. Moreover, this study shows that the numerically predicted friction reduction, depicted in Figure 6, could be used to acquire a ballpark figure on the potential friction reduction if the surface roughness, dimple geometry, contact geometry and load are in a reasonably similar range as those input to the numerical model.

The crank angle resolved friction traces from the experiment show that the shape of the textured cylinder liner friction trace is flatter around mid-stroke, especially for the tests at lower speeds. This indicates that the lubrication regime is affected. Typically, a flatter shape on the friction trace around mid-stroke is found at low speeds where the lubrication regime is more towards mixed or boundary; however, in this situation, this is not the case. Since it is known that the only difference between the two different cylinder liner types is the surface texture, the friction reduction most likely comes from the globally increased film thickness discussed previously. The shape of the friction traces for the tests with the reference liners, where friction increases with increasing speed, is sinusoidal-like because the film thickness cannot increase at the same rate as the speed is increased towards mid-stroke. This leads to increased sheer of the lubricant and results in greater friction force. In the tests with the textured cylinder liners, the enforced mean film thickness in the contact caused by the dimple geometry results in a much lower viscous shear of the lubricant. Because of the lower viscous shear, the crank angle resolved friction trace for the textured liners have a flat shape around mid-stroke. One could say that the surface texture acts as an “artificial mean oil film thickness enhancer”, which is the main cause of the large friction reduction from the surface texture.

It is also of great interest to investigate the possible influence on fuel consumption from the surface texture. The influence on fuel consumption from friction can be quantified by the friction power loss, which can be calculated by multiplying the speed with the friction force. Because the speed is greater around mid-stroke, this is the portion of the stroke that will most influence fuel consumption. Friction power is calculated by multiplying the measured friction force with the sliding speed at each sampled data point. First, the mean values of the friction power for each sampled piston cycle were calculated, then the average of the cycles from the second half of the test duration is taken as the average value of friction power for each test condition. This average friction power is depicted in Figure 11 as a function of equivalent truck engine speed to visualise the influence expected for an actual truck engine at various operating speeds. The potential for reduction of fuel consumption can clearly be seen. It can also be seen that the slopes of the curves are slightly less steep for the textured cylinder liners compared to the non-textured cylinder liners. That is, the absolute value of friction power reduction is greater with increased engine speed. This could be explained by a lower amount of mixed lubrication with increasing speed resulting in a less local increase in mixed friction on the plateau and a more beneficial effect from the artificially increased mean film thickness reducing viscous losses. However, the relative friction power reduction from the textured cylinder liners compared to the non-textured cylinder liner was similar for all three tested speeds. This is shown in Table 3 together with tabulated results from Figure 11. It should be noted that there might be a similar trend with a lower reduction at lower speeds also for the relative friction power reduction; however, additional speeds need to be tested before any conclusions on this can be made. This indicates that, if operating mainly in hydrodynamic lubrication, as in this study, the textured surface has a fixed relative effect on friction power that is most likely dependent on the amount of surface covered with dimples i.e., area density. It is reasonable to expect that a greater area density would contribute to an even larger reduction in frictional losses, as was also shown in the numerical study [9].

## 4. Conclusions

Friction in the piston ring–cylinder liner contact can be significantly reduced by surface texture. The surface texture was of dimple type with a diameter and depth of 300 µm and 3 µm respectively, covering 33% of the mid-section of the cylinder liner. The measured reduction in friction power loss was approximately 17% for all three different operating conditions tested. The piston rings used for the tests were of a standard type of twin land oil control ring used in most heavy-duty diesel engines. If the textured liner shows similar results in a fired engine, this amount of friction reduction will result in a measurable reduction in fuel consumption.

Moreover, the dimple geometry investigated in this work was derived from a numerical model developed in [9]. The friction reduction predicted by the numerical model was qualitatively compared to the measured friction reduction. The comparison showed a great correlation between the numerical and experimental results, indicating that the model prediction is accurate for this specific application and conditions.

## Figures and Tables

**Figure 1 materials-16-00665-f001:**
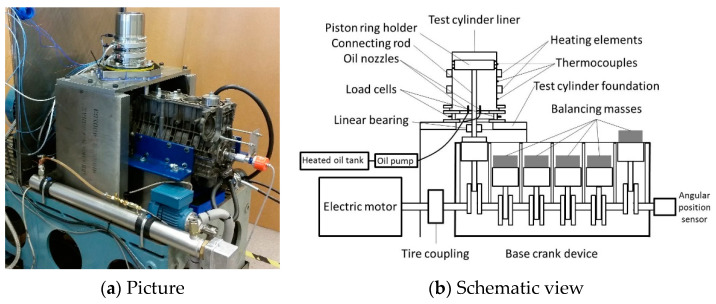
Experimental test equipment.

**Figure 2 materials-16-00665-f002:**
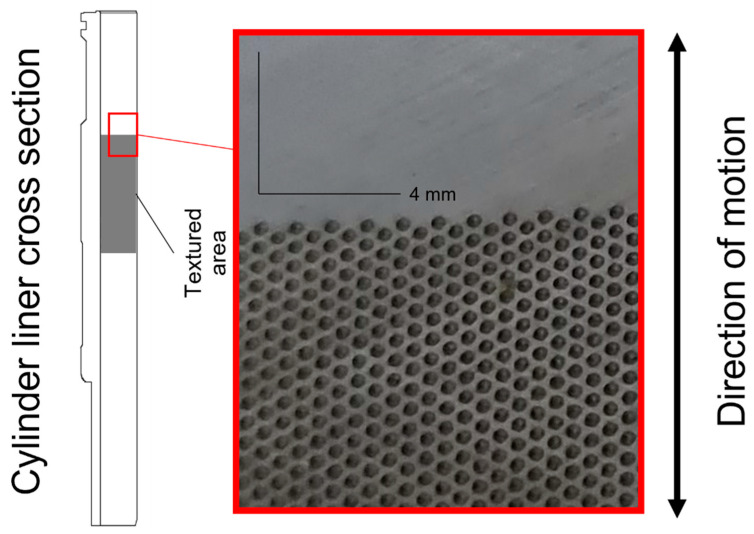
Schematic view of textured placement on the liner together with a picture of the cylinder liner surface with texture in a Zig-zag pattern.

**Figure 3 materials-16-00665-f003:**
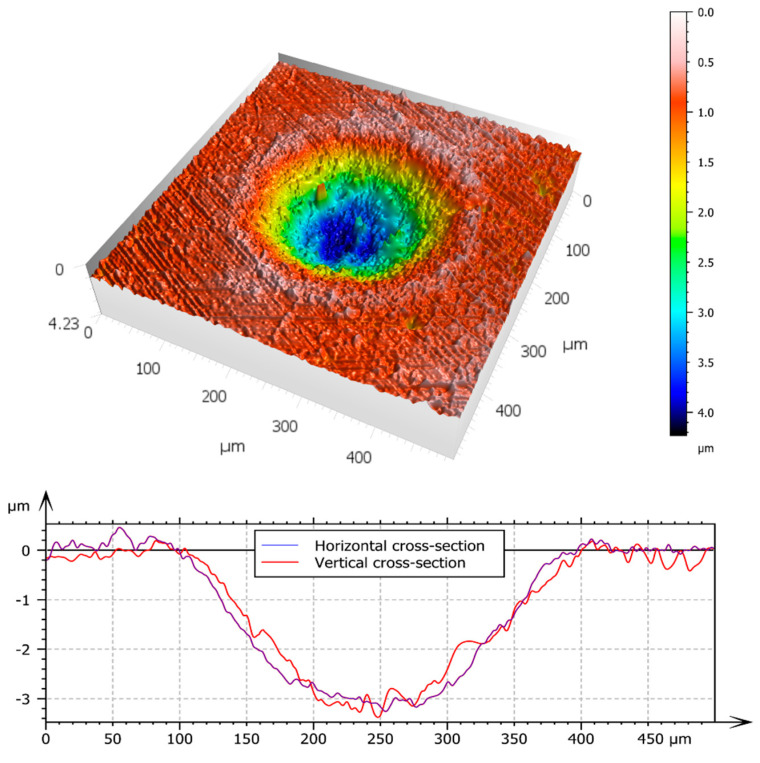
Measurement of a single dimple. 3D view as well as profiles in the horizontal and vertical directions.

**Figure 4 materials-16-00665-f004:**
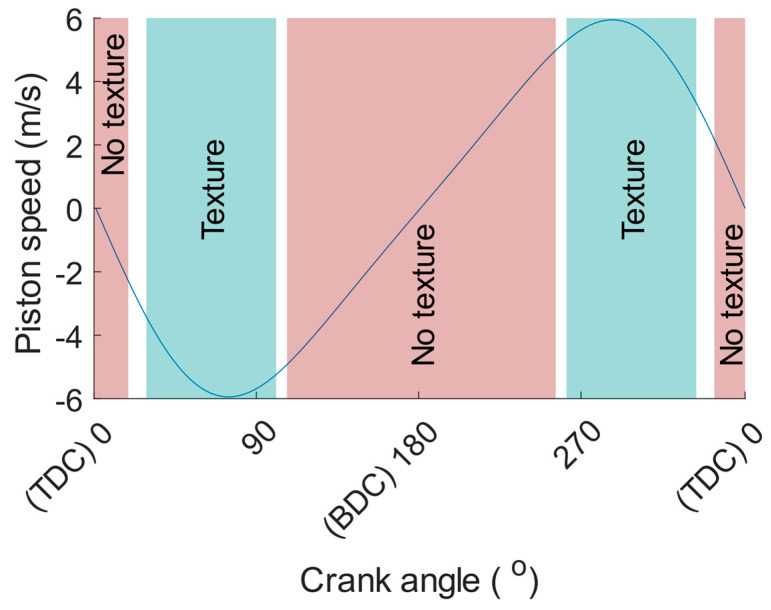
Piston ring speed as a function of crank angle degree at 1200 RPM illustrating when the oil control ring is completely over textured and non-textured areas on the cylinder liner. The non-coloured area between the two zones represents entering or leaving the textured area on the cylinder liner.

**Figure 5 materials-16-00665-f005:**
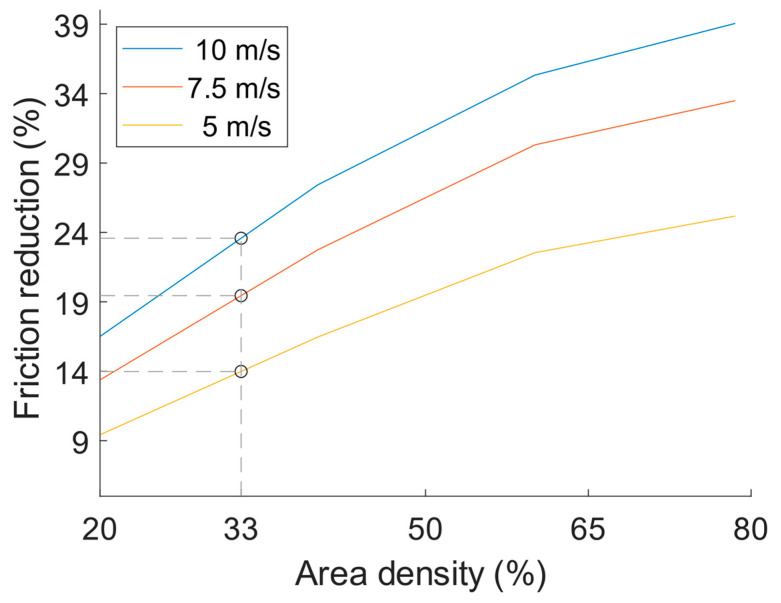
Friction reduction potential from a textured surface compared to a typical cylinder liner surface [9].

**Figure 6 materials-16-00665-f006:**
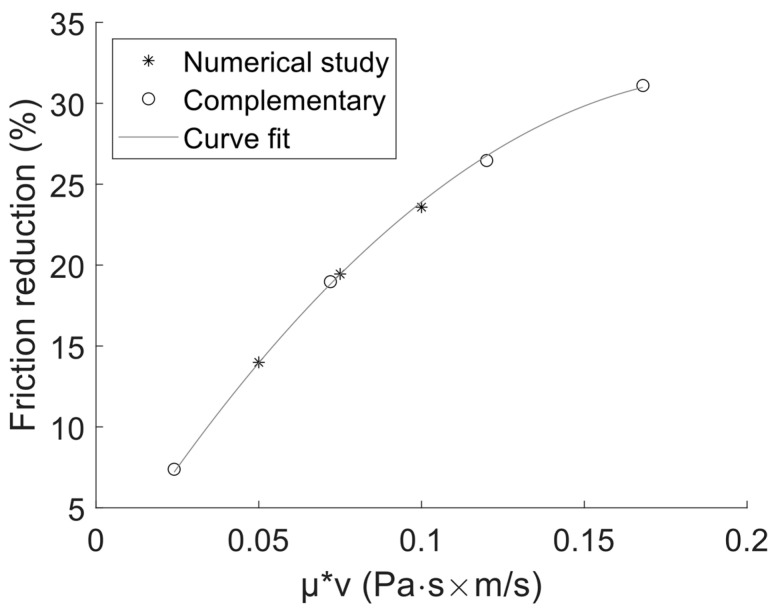
Friction reduction from surface texture predicted by the numerical model as a function of the product of viscosity and sliding speed.

**Figure 7 materials-16-00665-f007:**
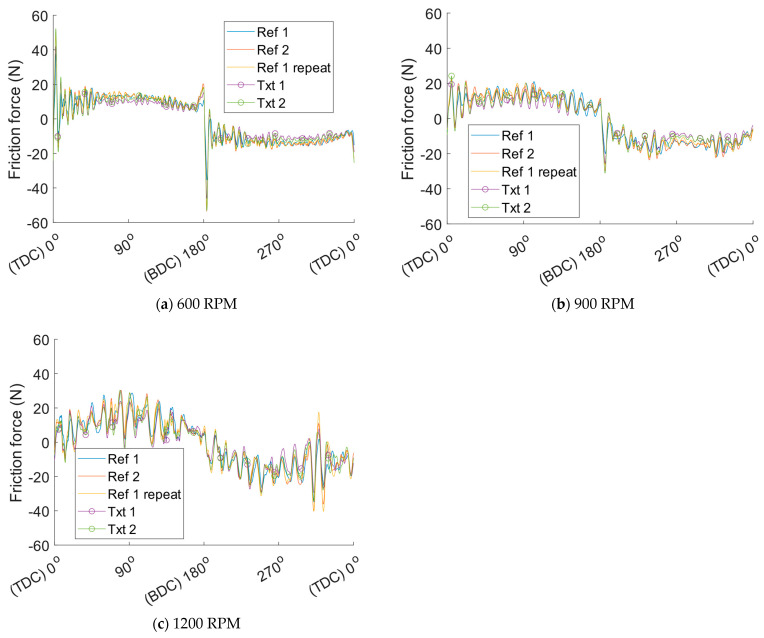
Crank angle resolved sampled friction force for the different reference and textured cylinder liners. Ref 1 repeat refers to the final repeated test performed with cylinder liner ref 1 to ensure the repeatability of the test method.

**Figure 8 materials-16-00665-f008:**
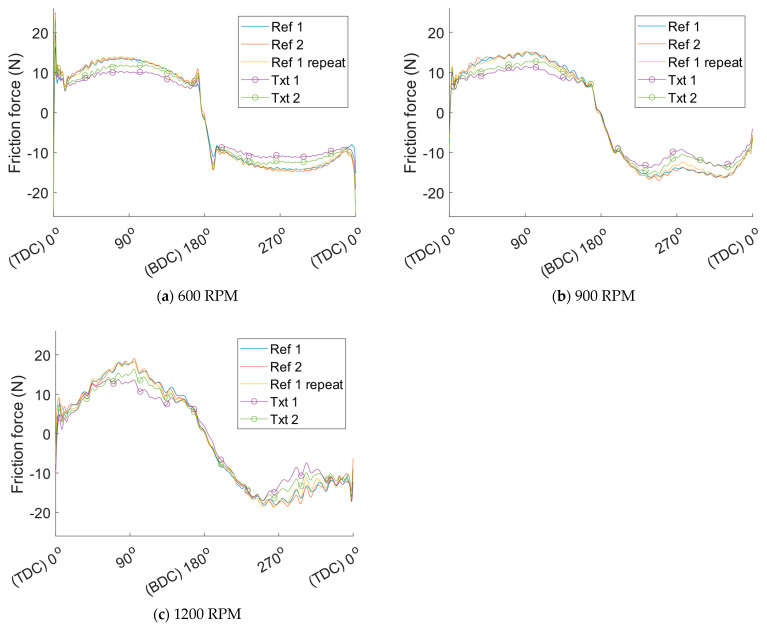
Crank angle resolved filtered friction force for the different reference and textured cylinder liners.

**Figure 9 materials-16-00665-f009:**
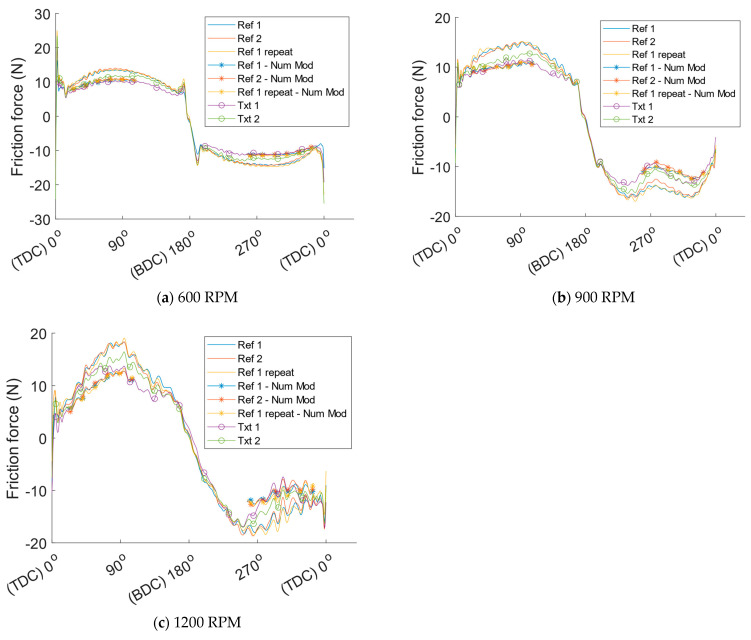
Friction reduction predicted by the numerical model, entitled “Num Mod”, compared to the measured friction force for all tested liners as a function of crank angle.

**Figure 10 materials-16-00665-f010:**
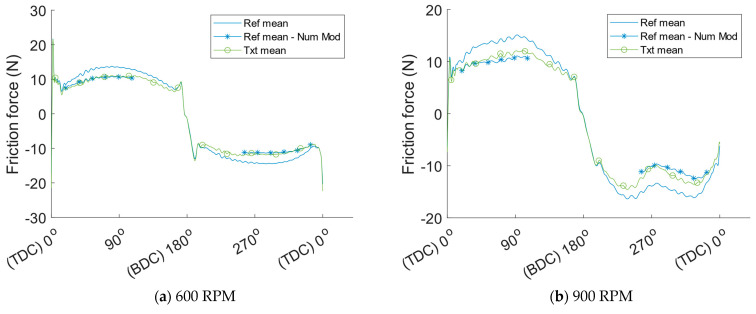
Mean value for all tested examples of each liner type compared to friction reduction predicted by the numerical model as a function of crank angle.

**Figure 11 materials-16-00665-f011:**
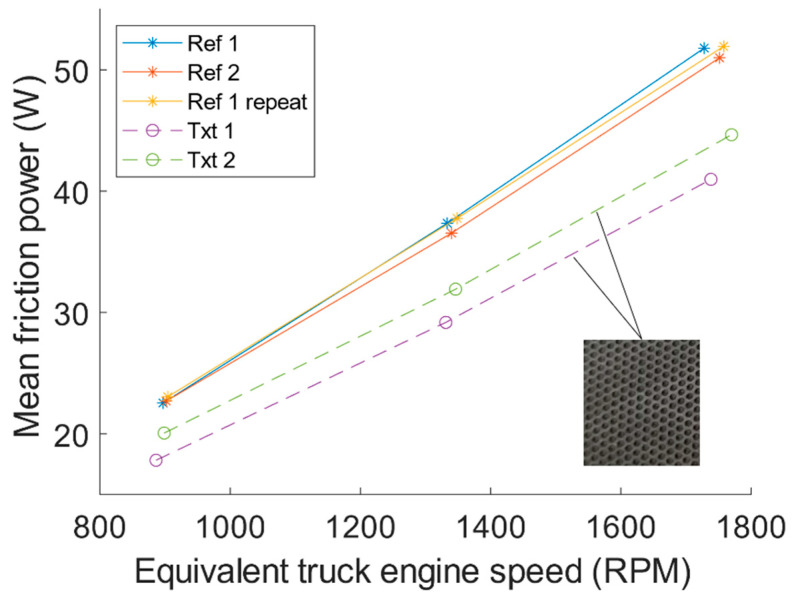
Mean friction power as a function of equivalent truck engine speed.

**Table 1 materials-16-00665-t001:** Tested cylinder liners.

Test ID (Order)	Cylinder Liner	Short Name
1	Reference 1	Ref 1
2	Texture 1	Txt 1
3	Texture 2	Txt 2
4	Reference 2	Ref 2
5	Reference 1	Ref 1 repeat

**Table 2 materials-16-00665-t002:** Design of experiment (DOE).

Test Condition	Rig Speed (RPM)	Rig Temperature (°C)	Equivalent Truck Engine Speed (RPM)
1	600	61	1000
2	900	61	1500
3	1200	61	2100

**Table 3 materials-16-00665-t003:** Measured friction power with mean values for each cylinder liner type and average friction reduction by the surface texture of this investigation.

Equivalent Truck Engine Speed	Friction Power (W)							Friction Power Reduction (%)
	Ref 1	Ref 2	Ref 1 Repeat	Txt 1	Txt 2	Mean Ref	Mean Txt	
900 RPM	22.5	22.7	23	17.8	20.1	22.7	19	16.3
1340 RPM	37.3	36.5	37.8	29.2	31.9	37.2	30.6	17.7
1750 RPM	51.8	51	51.9	41	44.6	51.6	42.8	17.1

## Data Availability

The data presented in this study are available on request from the corresponding author. The data are not publicly available due to [insert reason here].

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
