# Peer review of "Friction Reduction by Dimple Type Textured Cylinder Liners—An Experimental Investigation"

_materials, 2023, doi:10.3390/ma16020665_

Round 1
Reviewer 1 Report
1. Abstract is really abstract, it should have qualitative and quantitative expression.
2. All images are given in tabular form and should be modified as required by the journal.
3. The text in all figures should be the same size and one smaller than the text in the paper.
4. The format of the table should be uniform in the entire text, not in different formats.
5. As shown in Fig. 6-9, there is little difference in different conditions. The enlarged graph should be set in the curve, and the deviation between the predicted value and the actual value should be stated, and the reasons should be explained.
6. Should the paper have a "Conclusion" section?
Reviewer 2 Report
The authors experimentally investigated the effect of texture on frictional losses between piston ring and cylinder liner. Experiments are performed with sliding speeds close to real piston sliding speeds. Qualitatively comparing the experimental results to a numerical model shows good correlation.
It has been decades for the investigation of surface texturing for tribological applications. It has been found under some circumstances, textured surfaces can lead to better lubrication and wear-resistance performances. However, the studies of the influence of textured surfaces on the frictional properties of real cylinder liners are still insufficient. Hence, this study is valuable for the engineering application of textured surface. The conclusions can be supported by the results presented in the manuscript. However, before the acceptance of the manuscript, there are still some issues should be addressed:
1. The literature review of surface texturing for tribological applications is insufficient. I suggest the authors to add some literatures in the introduction section to review the tribological application of surface texture more detailly. (https://doi.org/10.1016/S0301-679X(03)00104-X; https://doi.org/10.1016/j.carbon.2020.05.041; https://doi.org/10.1016/j.wear.2011.04.003; )
2. All the abbreviation should be defined before use.
3. The novelty of this study should more clearly stated in the abstract.
Reviewer 3 Report
Problem statement need to be improved.
Any reason why the speed cannot be the same to the real speed.
Please specify the standard the test is following.
Numerical model was not clearly describe in the paper.
Figure 1 not clearly shown the component. It is suggested to label photo in Figure (a) in numbering.
Better to add the cross-section view of the dimpled profile.
Figure 2. zig-zag pattern? not clearly seen the pattern.
It is good to have SEM images for the surfaces, before and after the test to observed the effect of the proposed profile. Comparison with other methods or profile also give value added to the paper.
No conclusion?
References are updated
Round 2
Reviewer 1 Report
The authors have modified the manuscript according to my comments. It could be published at present version.
Author Response
Thank you for your input.
Reviewer 3 Report
The authors addressed most of the comments given. However, I need further explanation about;
1. Why the title and the content is not tally? Title seems to just cover the experiment only but the content included the numerical as well. Better to focus on experimental and no need to included Ref in conclusion for the sake of comparison.
2. Figure 1 not label completely i.e. the Picture.
3. Zig-zag pattern not well described.
